# Influence of the El Niño Phenomenon on Shoreline Evolution. Case Study: Callao Bay, Perú

**Emanuel Guzman** [1,*], **Carmela Ramos** [1] **and Ali Dastgheib** [2]

1   Facultad de Ingeniería Civil, Universidad Peruana de Ciencias Aplicadas (UPC), Avenida General Salaverry 2255 San Isidro, Lima 27, Peru; pccicram@upc.edu.pe
2   Coastal systems engineering and port development, IHE-Delft Institute for Water Education, Westvest 7, 2611 AX Delft, The Netherlands; a.dastgheib@un-ihe.org
*   Correspondence: pcipeguz@upc.edu.pe; Tel.: +511975130526

**Abstract:** Analyzing the long-term behavior of the coastlines in the vicinity of river mouths and estuaries usually relies on the mean (predicted) values of the sediment discharge from the river. However, this approach does not consider low frequency, severe events, such as El Niño (EN), that can have a large effect on coastlines. While the effects of an EN on flooding and droughts are well studied, little information exists about its effects on coastal zones, and especially on the evolution of coastlines. In early 2017, an EN occurred in the equatorial Pacific Ocean, and the country of Peru was affected with high precipitation levels, and extreme river discharges and flooding. During this event, in the district of Lima, the Rimac River discharged a huge amount of sediment into the Callao Bay, and the shoreline accreted approximately 1 km, demonstrating the significant effects that an EN can have on coastal zones. To explore these effects, this paper studies the influence of an EN on shoreline evolution in the Callao Bay by analyzing Landsat images from 1985–2019 to understand the shoreline evolution and identify changes to the coastline. Results show that when an extraordinary EN occurs (e.g., 1982–1983, 1997–1998, and 2017), the shoreline experiences high accretion compared to when a smaller, or no EN occurs. During these events, a significant delta forms at the south end of the bay, and the redistribution of the accumulated sediment by wave action causes the accretion of the adjacent coastlines for as far as 7 km north of the river mouth. This shows the importance of these events for the wellbeing of coastlines adjacent to river mouths affected by EN.

**Keywords:** shoreline evolution; EL NIÑO and coastal process; coastal process

## 1. Introduction

Analyzing the long-term behavior of coastlines in the vicinity of river mouths and estuaries usually relies on the mean (predicted) values of sediment discharge from the river to the coast [1,2]. However, this approach does not consider low frequency, severe events that can have a large effect on the coastline, even changing its evolution, and therefore affecting the environmental, commercial, and social interests of the region. One of these rare events with high impact is El Niño (EN). An EN is a recurrent phenomenon that affects countries around the Pacific Ocean, characterized by an anomalous increase of the sea surface temperature (SST), as well as an increase in the sea level in the central-eastern equatorial Pacific. These ocean effects can cause an increase in precipitation on land that overwhelms natural catchment systems, raises river water levels, inundates flood plains, and eventually results in abnormally large discharges of sediment into the coastal zones at the estuary. Specifically, the arid regions of Peru and Chile receive a large increase of precipitation during a strong EN, causing the floods and river inundations just described [3]. The effects of an EN on the catchment processes of precipitation, alluvial flooding, and drought are well studied, but less information exists regarding

the secondary effects of an EN on coastal morphological evolution and coastline variation. Existing studies demonstrate that some coastal zones and their shorelines, specifically in the western United States, experience significant changes during EN. For example, the shoreline of the western coast of the U.S. retreated around 23% more during the EN years of 2009–2010 than during normal conditions, with beaches severely eroded due to changes in wave direction [4]. On the central coast of California, increased wave heights, sea level, and cyclone activities due to an EN have also affected shoreline variations [5]. In the EN winter of 2015–2016 on the California coast, a high wave energy flux, coupled with elevated water levels, caused unprecedented levels of shoreline retreat [6]. Between 1997 and 2000 on the Pacific Northwest coast of the U.S., six major storms occurred, generating deep-water wave heights greater than 10 m [7]. These conditions could affect sediment transport in coastal zones that could increase the erosion or accretion of beaches. Thus, any increase in the frequency of extreme EN, and its related La Niña, will have a considerable effect on populated coastal zones of the Pacific Ocean due to extreme coastal erosion and flooding [8].

In the Peruvian context, the northern and central coasts are the areas most affected. During the EN in 1997–1998, precipitation in the northern coast increased to 30 times its normal range, causing a total accumulated rainfall of 1802 mm [9]. The recent EN event in 2017, called "El Niño Costero", occurred between January and May 2017, and was characterized by a rapid development and large impact, localized in the northwest of South America [10]. The precipitation levels that occurred in this period in Peru are similar to those in the strongest EN events that occurred in 1982–1983 and 1997–1998 [11]. Although these events greatly affected the Peruvian coastline, they have not been quantified, analyzed, or studied in-depth.

It should also be mentioned that some evidence suggests that in extraordinary EN events, such as in 1997–1998, wave climate along the Peruvian coast is slightly different than normal years [12,13].

CENEPRED (Centro de Estimación, Prevención y Reducción de Riesgos y Desastres), the official Peruvian institution for the prediction and prevention of disasters, provides information on the ENs that have occurred between 1951 and 2009, and report that the two extraordinary ENs of 1982–1983 and 1997–1998 had an extreme impact on human and economic activities, especially in the fishing industry [14], and stand as the two strongest on record. There is also some evidence that the third strongest EN in the twentieth century occurred in 1925, before CENEPRED started tracking disasters, due to the high human and economic losses associated with its severe rainfall and flooding of coastal zones [15].

Using the available satellite imagery from Landsat from the years 1985–2019, this study will focus on the influence of these EN events on the coastline evolution of the Callao Bay in Peru.

## 2. Case Study

The Callao Bay is located on the central coast of Peru, within the Lima metropolitan area, and belongs to the Constitutional province of Callao (Figure 1). It serves as one of the most important Peruvian bays due the economic activities related to Peru's biggest port, the Terminal Port of Callao (TP-Callao), located to the south of the Rimac River estuary (Figure 1). TP-Callao has two breakwaters approximately 1.3 km long, protecting the areas south of the river mouth that include fishing and other forms of economic industry, as well as recreational and recently developed tourism activities.

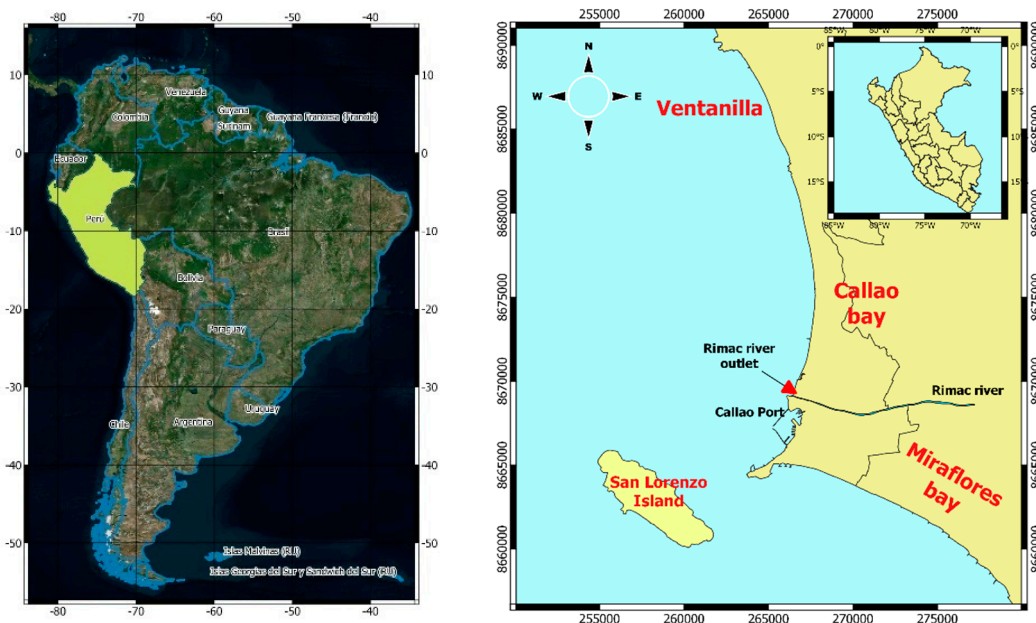

**Figure 1.** Study area.

The offshore waves in front of Callao Bay have a mainly Southwest direction (Figure 2), however due to refraction and diffraction effects from San Lorenzo Island situated in front of Callao Bay, the incident waves generated have a predominantly Westerly direction and their mean wave height diminishes from 2.1 offshore to 0.5 m at the bay coastline [16].

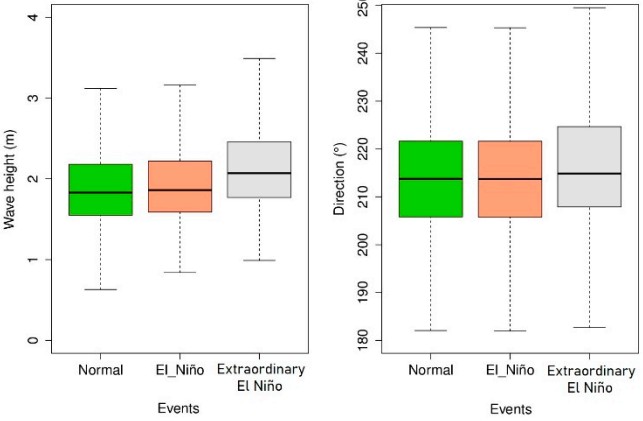

**Figure 2.** Comparison of wave climate conditions in normal, El Niño, and extraordinary El Niño years. Figure was created based on data presented in [13].

The tidal regime in Callao Bay is semidiurnal, with M2 as its major constituent (Table 1), with 0.22 m of amplitude [17]. Little information exists concerning the bay's sea bottom sediment, and it does not present results of an analysis of sediment grain-size in time and space. A 2011 study found that mean sediment grain-size in the bay is within the range of 0.85–5.0 mm, with higher values on the northern side of the Rimac River mouth [18]. A geologic map by the Regional Government of Callao (Gobierno Regional del Callao, or GORE-Callao) in 2011 also indicates the shoreline of Callao Bay as a sandy beach [19].

**Table 1.** Main astronomical constituents for the tidal regime in Callao Bay extracted from TPXO 7.2 model data [17]. (geographical coodinates: −77.5°, −12°).

| Name | Amplitude (m) | Phase (°) |
|------|---------------|-----------|
| M2 | 0.22 | 314.8 |
| S2 | 0.08 | 323.5 |
| N2 | 0.06 | 282.2 |
| K2 | 0.03 | 331.0 |
| K1 | 0.14 | 28.3 |
| O1 | 0.07 | 347.9 |

The Rimac River mouth is located in the middle of Callao Bay, and it is the main source of sediment to the bay area (Figure 2). Its basin has a surface area of 3300 km² and 160 km of length [20]. One of the most important roles of the Rimac River is to provide 80% of the current domestic and commercial water usage in the City of Lima [21], with the summer season showing higher discharges at the river mouth when compared to the rest of the year. Another important role of the Rimac River is to provide sediment that maintains coastline equilibrium. In extreme events associated with an EN phenomenon, the Rimac River has flooded some parts of the Lima, and inland Chosica, districts, causing a loss of property and human life. These extreme events not only cause flooding due to increased discharge of the river, but can change the sediment load of the river as well, resulting in a significant change to the coastal morphology around the river mouth.

Normal Rimac River discharges begin to increase from January until May, reaching their maximum in mid-March [22,23]. EN events follow this timeline but strongly increase water and discharge levels (Figure 3). For example, during the EN in 2017, discharge reached a maximum of 130 m³/s, which is two times the average historical value (Figure 3b). It is also shown that during extreme EN events, suspended sediment yield (SSY) of rivers in the Peruvian Andes increase to 30–60 times their average level of normal years [24]. During these events, 82%–97% of SSY occurs between January and April, which coincides with the summer season in the southern hemisphere [24]. Therefore, the Rimac River provides a higher sediment supply during the summer seasons between January and April, and especially during those EN events where discharges increase significantly over normal conditions.

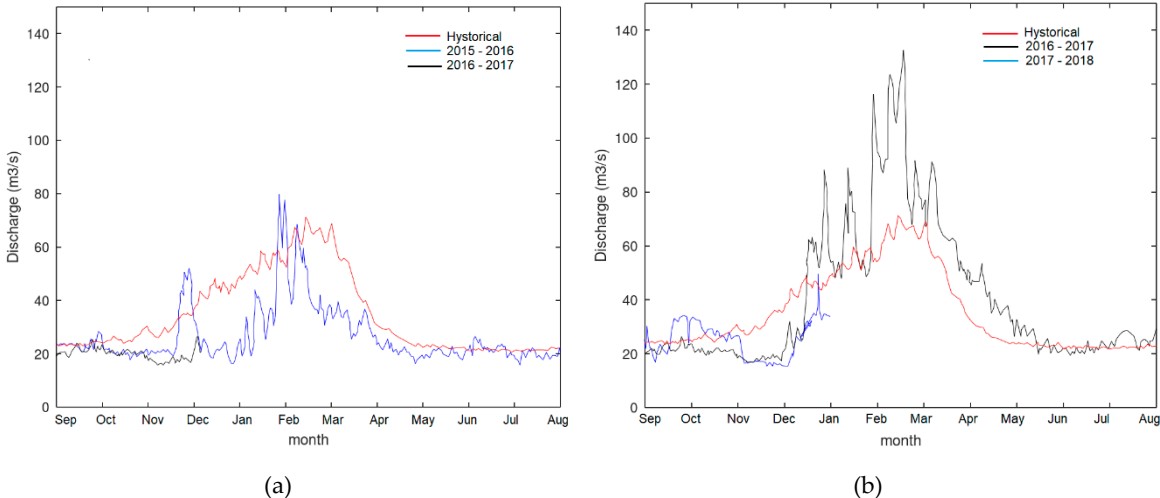

| (a) | (b) |

**Figure 3.** Historical and El Niño discharges of the Rimac River. (**a**) Discharges in a normal year from September 2015 to January 2016. (**b**) Discharges when an El Niño (EN) occurred from September 2016 to February 2018. Figure was reproduced from ENFEN (Estudio Nacional del fenómeno EL NIÑO) public domains reports [22,23].

Previous studies indicate that in 2017, during what was named the "El Niño Costero", the Rimac River discharged around 3.4 million cubic meters of sand between January and March [16]. This volume

was estimated by comparing bathymetries between 2013 and 2017 in a surveyed area around 2.9 km along the shoreline and 1.2 km in a cross-shore direction, and shows that all the sediment accumulated in the area is mainly due to Rimac River discharge. It was also shown that the study area has a littoral transport of around 30,000 m$^3$/year towards the north [16].

## 3. Materials and Methods

Satellite images from Landsat for the dates 1985–2019 (Table 2) were obtained from the United States Geological Survey (USGS) Earth explorer home page (https://earthexplorer.usgs.gov/) [25]. The resolution of Landsat 4 and 7 images are 30 m per pixel, while the resolution of Landsat 8 images is 15 m per pixel. In both cases RMSE (Root Mean Square Error) is less than 12 m and the margin of error is around 7–15 m for Landsat 8 and 7 respectively. To analyze shoreline variations, open source software was employed. Satellite images were processed using QGIS software to make the band combinations and obtain coordinates of the shoreline, while the processing and interpolation analyses of shorelines were carried out by Octave. All images were processed using band combination 564 (land/sea) to identify land and water bodies.

Figure 4 presents the methodology used in the study, and the steps of their implementation are explained as follows:

- Landsat images are downloaded from the Earth Explorer website. Images were selected from 1985 to 2019 from Landsat 4 to Landsat 8, considering cloud coverage in the study area.
- The shoreline was digitized from selected satellite images. Band combination was used to differentiate the coast from the sea.
- A new origin of coordinates (Figure 5) was defined in x = 266703 and y = 8668120 (UTM zone 18S) with a 74° rotated angle. In this step new shoreline coordinates were obtained.
- A 2D interpolation is executed along the new coordinate system to calibrate all points to the same reference level.
- Distances are calculated with respect to the previous year to determine positive and negative variations of the shoreline, and the rate of variations is measured in m/month.
- A 2D matrix in time and space is constructed to present trends. This analysis allows for the identification of coastal erosion/sedimentation zones.
- Over the 2D matrix, 4 reference points (Figure 5 and Table 3) are defined to track changes in area shoreline across time and calculate trends of shoreline evolution for each reference area.

**Table 2.** Landsat satellite image used in the analysis.

| Year | Month | Day | Year | Month | Day | Year | Month | Day |
|------|-------|-----|------|-------|-----|------|-------|-----|
| 1985 | 1 | 10 | 1996 | 4 | 14 | 2007 | 3 | 28 |
| 1986 | 5 | 5 | 1997 | 2 | 3 | 2008 | 3 | 21 |
| 1987 | 3 | 5 | 1997 | 12 | 13 | 2009 | 4 | 2 |
| 1987 | 12 | 18 | 1998 | 5 | 6 | 2010 | 10 | 30 |
| 1988 | 3 | 23 | 1998 | 12 | 16 | 2011 | 3 | 11 |
| 1988 | 4 | 8 | 1999 | 2 | 18 | 2015 | 5 | 5 |
| 1988 | 11 | 2 | 1999 | 6 | 26 | 2016 | 3 | 4 |
| 1989 | 12 | 23 | 1999 | 12 | 3 | 2016 | 12 | 17 |
| 1990 | 1 | 8 | 2000 | 12 | 5 | 2017 | 2 | 19 |
| 1991 | 2 | 28 | 2002 | 3 | 13 | 2017 | 4 | 15 |
| 1991 | 12 | 13 | 2002 | 5 | 9 | 2018 | 1 | 21 |
| 1993 | 2 | 17 | 2003 | 4 | 1 | 2018 | 4 | 18 |
| 1993 | 9 | 20 | 2004 | 1 | 31 | 2019 | 2 | 16 |
| 1993 | 10 | 7 | 2004 | 12 | 7 | | | |
| 1995 | 5 | 14 | 2006 | 4 | 26 | | | |

**Table 3.** Control point coordinates in the UTM system.

| Point | Coordinates (UTM 18S) | | Distance from New Origin Coordinates (m) | Location |
|---|---|---|---|---|
| | X | Y | | |
| P01 | 266981.0 | 8669064.0 | 1000 | Rimac river |
| P02 | 267311.0 | 8670021.0 | 2000 | Acapulco beach |
| P03 | 267640.0 | 8670970.0 | 3000 | Fertiza beach |
| P04 | 268568.0 | 8673880.0 | 6000 | Oquendo beach |

Main considerations of the methods are as follows:

- Tidal variations shown in the digitalization of coastline were neglected due to their very low range of around 0.5 m (Table 1). The maximum water level difference between the consecutive images used in the analyses was 0.2 m. Given the average beach slop of 0.01, this could lead to a maximum 20 m of inaccuracy in the horizontal position of a coastline. The maximum possible inaccuracy due to this simplification was smaller than the resolution of the images.
- Satellite images show the total variation in shoreline, including changes due to other processes of coastline variation, e.g., wave and tidal forcing as well as for sea level rise, which for Peru was 5 cm from 1942 to 2001 [26].
- Cubic spline interpolation was performed between digitalized points.

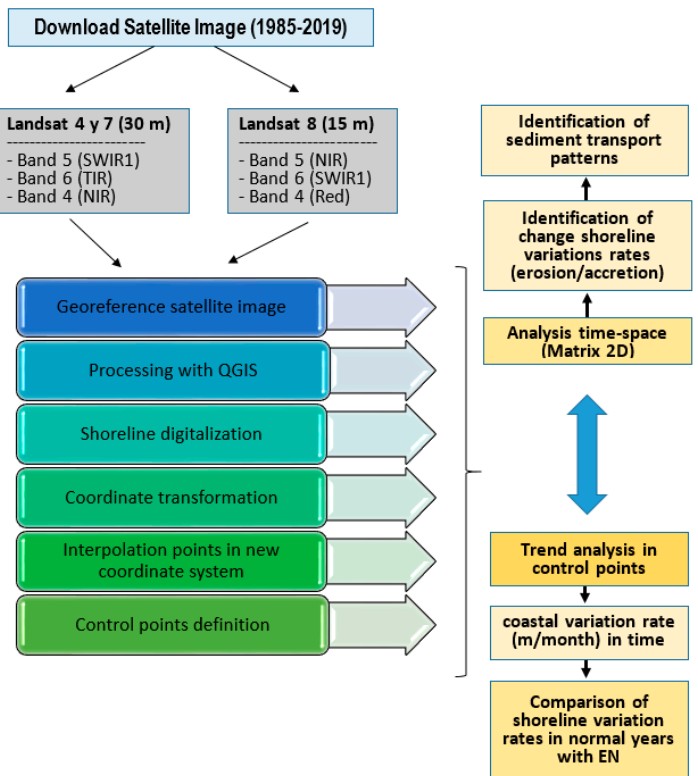

**Figure 4.** Scheme of the analysis of shoreline variation applied in this study.

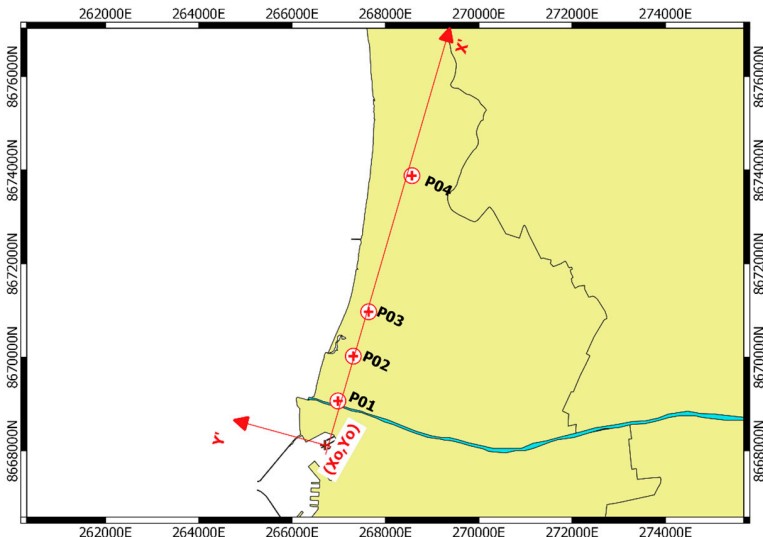

**Figure 5.** New coordinate system and control points defined in study area.

## 4. Results and Discussions

Landsat images in Figure 6 show first the shoreline of the Rimac River estuary in May 1998 after the EN in 1997–1998, where a 1700 m long by 750 m wide delta formed, and the estuary state in December 1998, eight months later. In December, the Rimac river delta still exists at the river mouth, but has been eroded by actions of longshore, wave-driven currents. Landsat images in Figure 7 show similar conditions occurred in EN 2017, when first a 1800 m long by 950 m wide delta formed in April 2017, and then its state in January 2018, 9 months later. Both cases confirm the previous studies, where the Rimac delta presents a clear northward movement, mainly due to wave driven sediment transport in the Callao Bay [16].

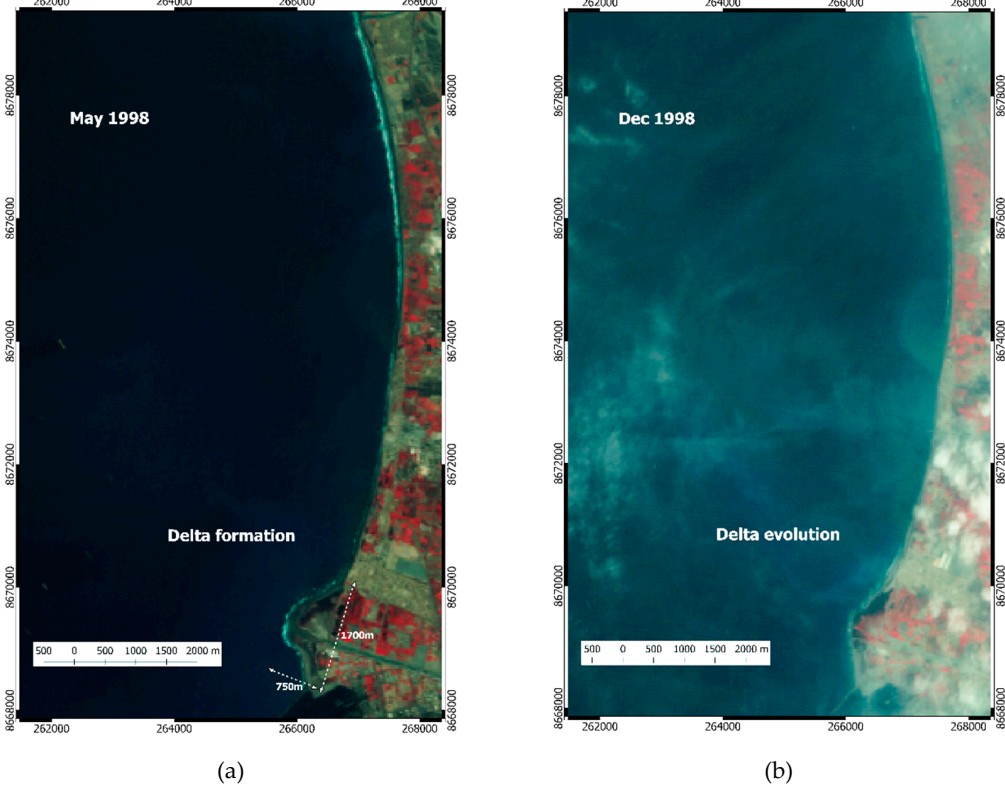

(a)                                                              (b)

**Figure 6.** Rimac Delta formation in 1998, (**a**) Just after the EN occurs, (**b**) 8 months later.

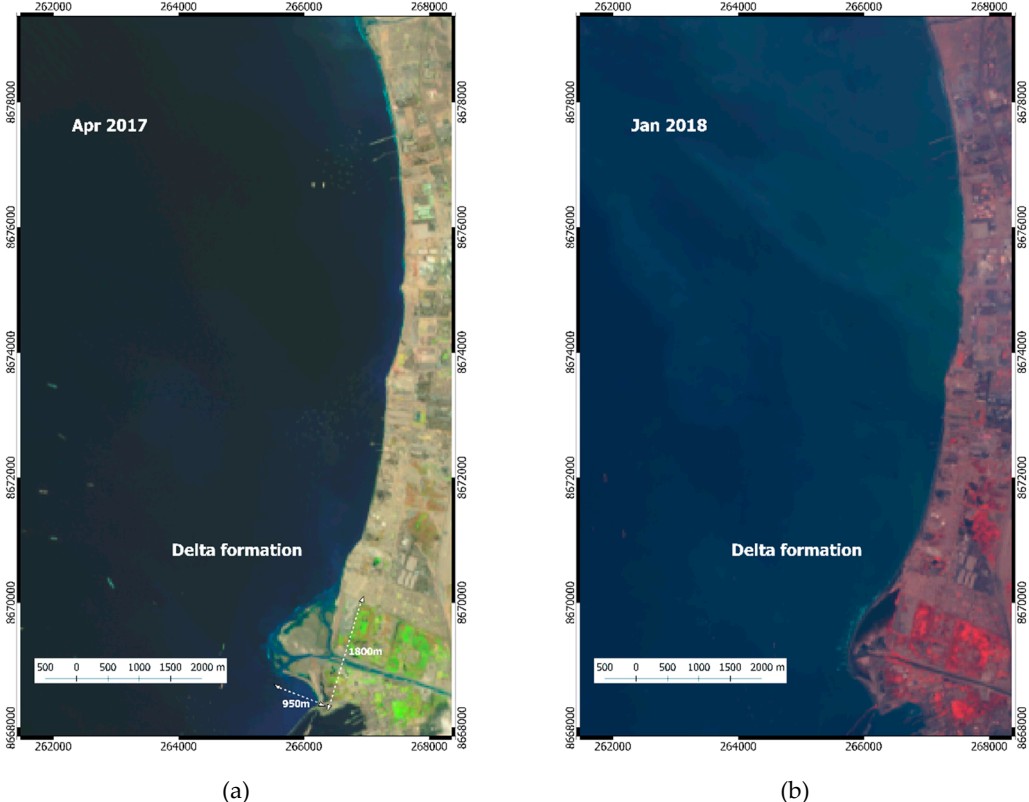

(a)                                    (b)

**Figure 7.** Rimac Delta formation in 2017, (**a**) Delta just after the EN occurs, (**b**) Delta 9 months later.

Comparisons of shoreline evolution between May and December 1998 (Figure 8), and between April 2017 and January 2018 (Figure 9), show similar behavior after delta formation. The Rimac Delta erodes around 250 m, and its sediment is transported to the north by the estimated annual rate of wave-driven sediment transport of 30,000 m³/year, accreting the adjacent coastline beaches by 30–70 m [16]. The tidal conditions presented in Table 1, classify in the microtidal range (<2 m) [27], with sediment transport dominated by wave-driven currents [28].

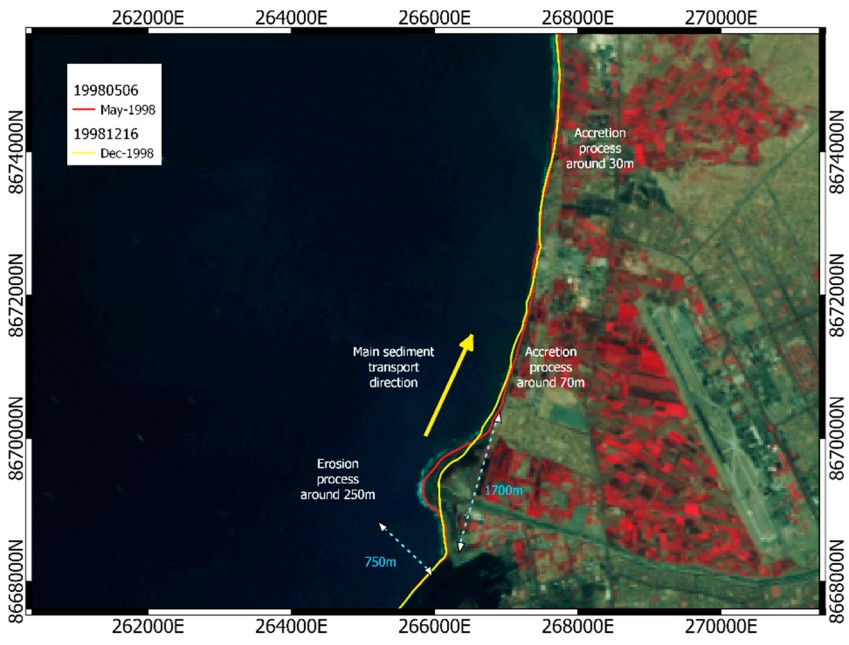

**Figure 8.** Shoreline evolution between May 1998 and December 1998.

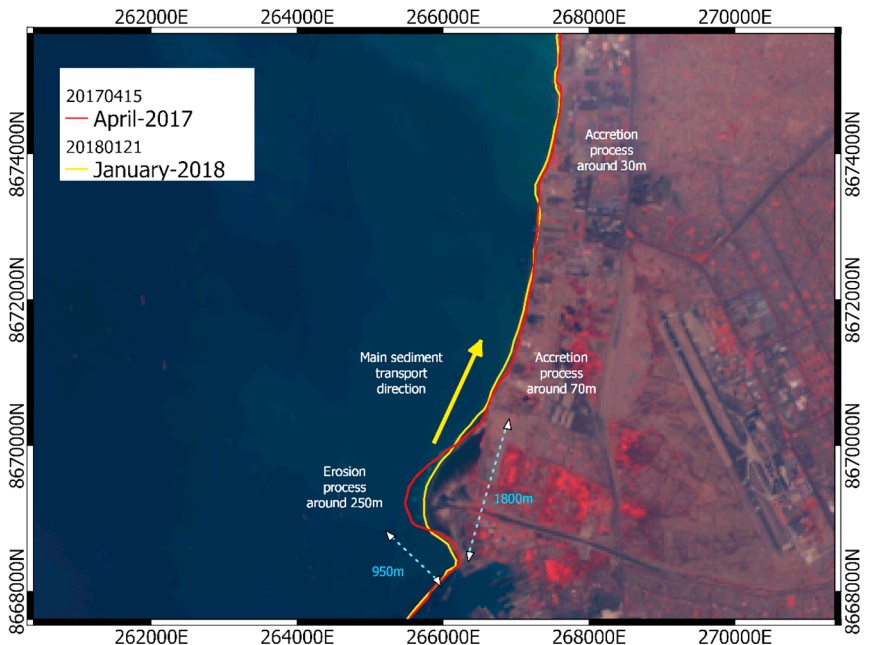

**Figure 9.** Shoreline evolution between April 2017 and January 2018.

But not all ENs had significantly influenced shoreline variations. Neither the ENs of 1986–1987 nor 1995 caused a delta formation at the mouth of the Rimac River (Figure 10). This could be due to differences in the magnitude of rainfall during those EN events, as the extraordinary ENs of 1982–1983, 1997–1998 and EN 2017 all registered the highest records of rainfall reported [11].

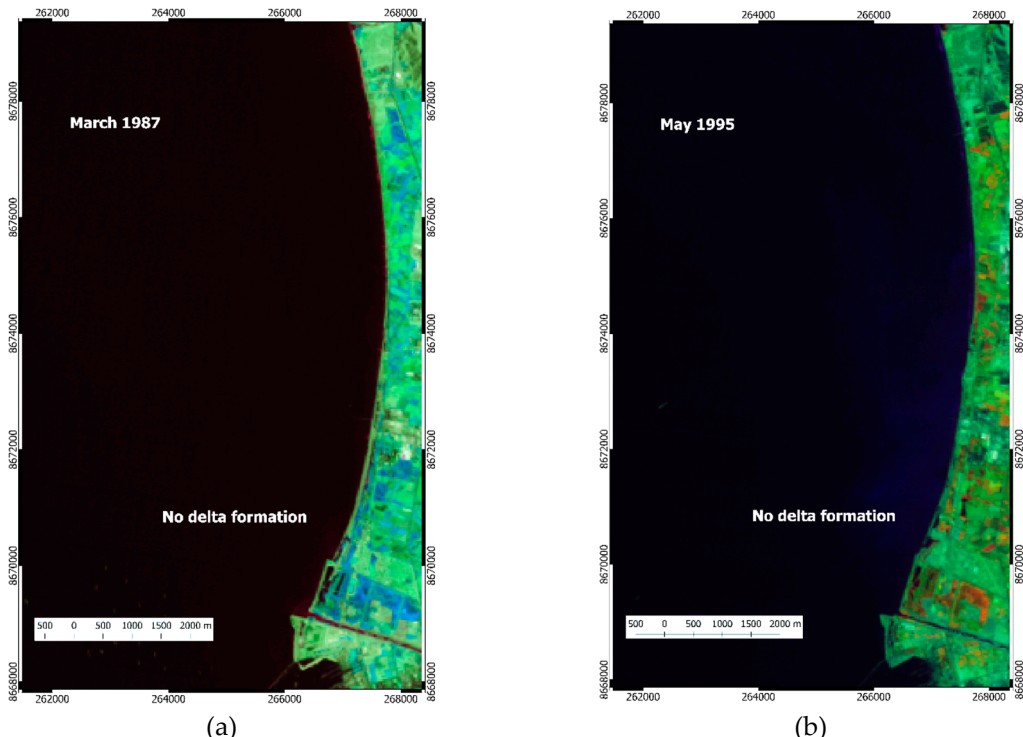

**Figure 10.** Rimac Delta evolution (**a**) during moderate EN in 1987, (**b**) during a weak EN in 1995.

To check the possible effect of EN on other coastal processes that can contribute to coastal evolution, detailed analyses of the wave climate was performed here. The deep-water wave climate at Callao Bay in normal years (1987 and 1995) shows similar conditions as during EN years (1998 and 2017), with only

slightly higher values of average wave height occurring in the extreme EN of 1998 (Figure 11a). Thus, an EN does not have a considerable effect on wave climate and the resulting sediment transport. In general terms, wave height increases between May and September, with maximum values occurring in July and August (Figure 11a). The 99 percentile of significant wave height also has a maximum value of 4.0 m between June and July (Figure 11b). Deep-water wave direction did not vary significantly in EN years (1998 and 2017) compared to normal years, with all cases presenting a main direction from the southwest (Figure 11c). These EN wave conditions in the Callao Bay are much less extreme than those found in other locations around the world, for instance in California, where greater than 10 m offshore wave heights in major storms have been reported during EN years [7].

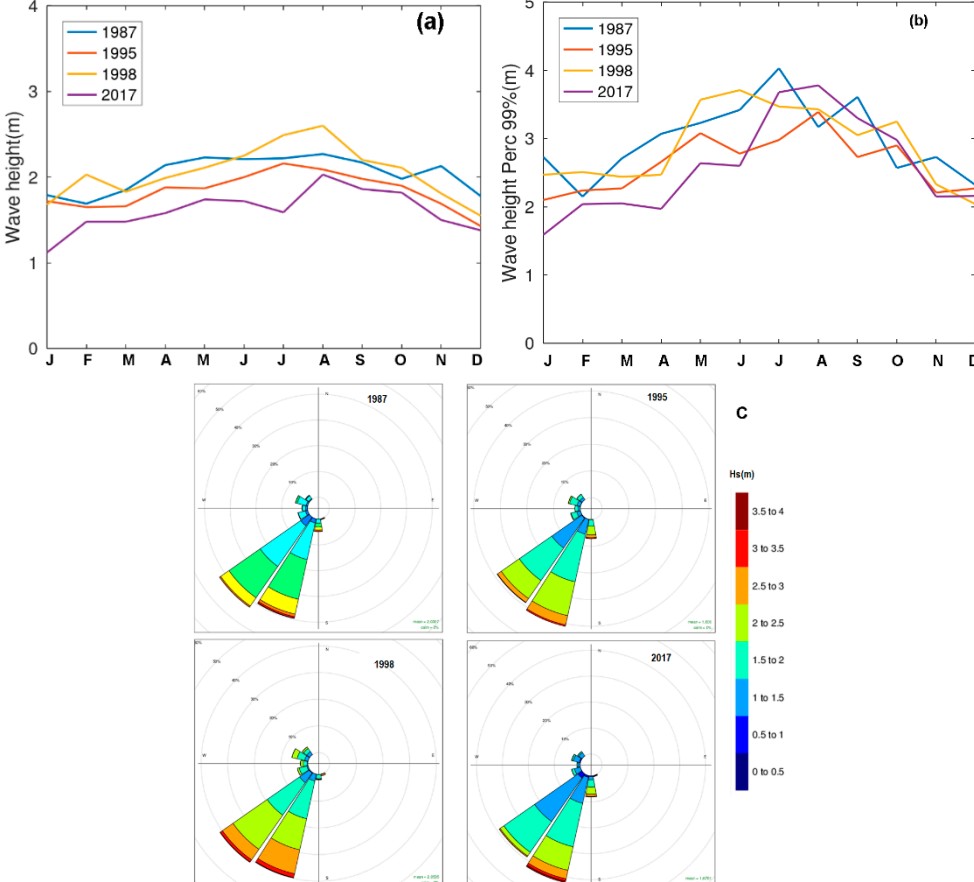

**Figure 11.** Wave height in deep-water of Callao Bay in 1987, 1995, 1998, and 2017. (**a**) Average significant wave height. (**b**) Percentile 99 of significant wave height. (**c**) Changes in wave height for normal years (1987 and 1995) and EN years (1998 and 2017).

A 2D space-time matrix was constructed to analyze the rates of accretion or erosion of the coastline in m/month across time and space, that considered all the details described previously for the shoreline coordinates obtained from the Landsat images (Figure 12). The X-axis represents the time that corresponds to the satellite time record, and the Y-axis represents the distance (in meters) from the original coordinate. The Rimac river mouth is located around 1000 m from the "x" origin coordinates. This matrix shows how the coastline changes by both the sedimentation and erosion processes over time (Figure 13). The results show that the highest shoreline variations occur close to the Rimac river mouth, and particularly during the 1993, 1998, and 2017 ENs, with an accretion rate in the range of 35–40 m/month. After this period, perceptible erosion occurred at similar rates. Additionally, in areas adjacent to the north of the river mouth, like Acapulco beach, the coastline had accreted within a few months of EN events, while the locations further north, e.g., Fertiza and Oquendo beaches, experienced the accretion with a delay of 2–3 years after EN events (Figure 13).

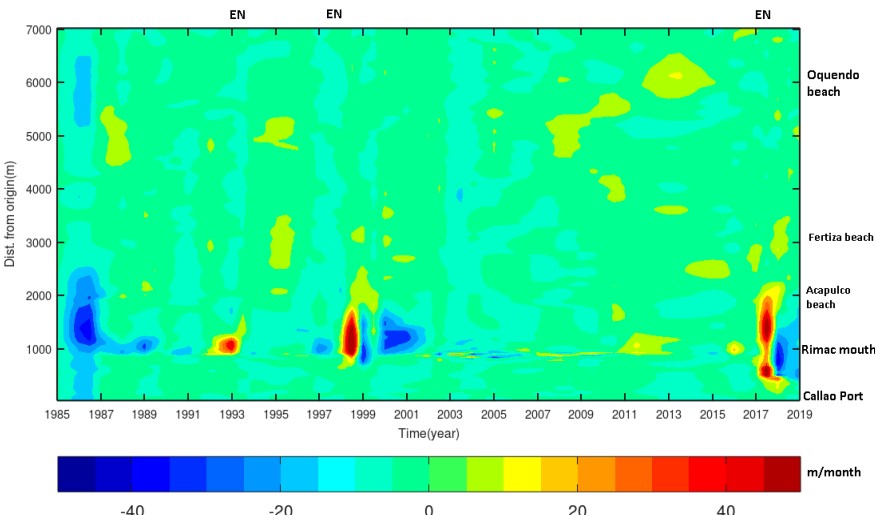

**Figure 12.** Temporal and space evolution of the Callao Bay shoreline. Figure shows shoreline variations in time and space, indicating EN events.

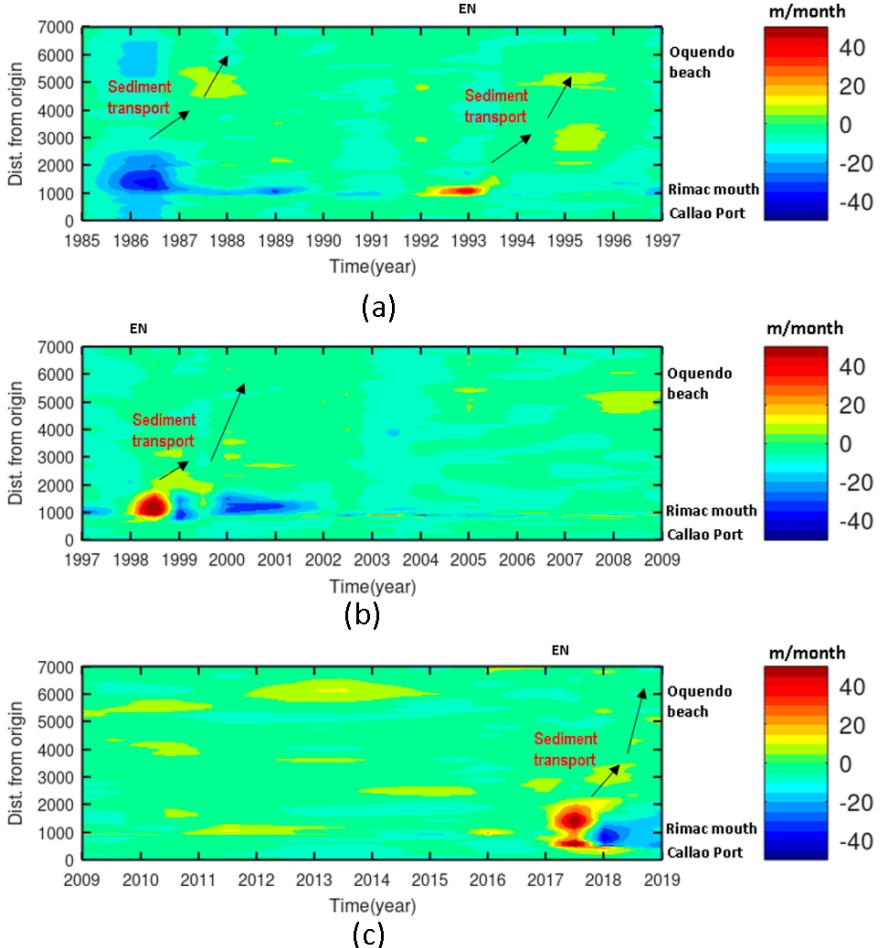

**Figure 13.** Detailed temporal and space evolution of the Callao Bay shoreline for EN events. Figure shows the shoreline variation-rate. (**a**) During EN effects in 1993, (**b**) During EN effects in 1997–1998, (**c**) During EN effects in 2017.

The linear regression results for shoreline variation (m/month/m) for each control point (P01, P02, P03, and P04) indicates that there existed a positive trend in the rate of shoreline variations, especially

for P01 and P02, which were closer to the river mouth, and show a rate of sedimentation of 0.275 and 0.209 m/month/m respectively (Figure 14). Likewise, P03 and P04 show a low positive trend, in the range of 0.067–0.15 m/month/m (Table 4).

**Table 4.** Linear regression coefficients in control points.

| Loc. | Slope (m/month/m) | Conclusion |
|------|-------------------|------------|
| P01 | 0.275 | Accretion |
| P02 | 0.209 | Accretion |
| P03 | 0.067 | Accretion |
| P04 | 0.150 | Accretion |

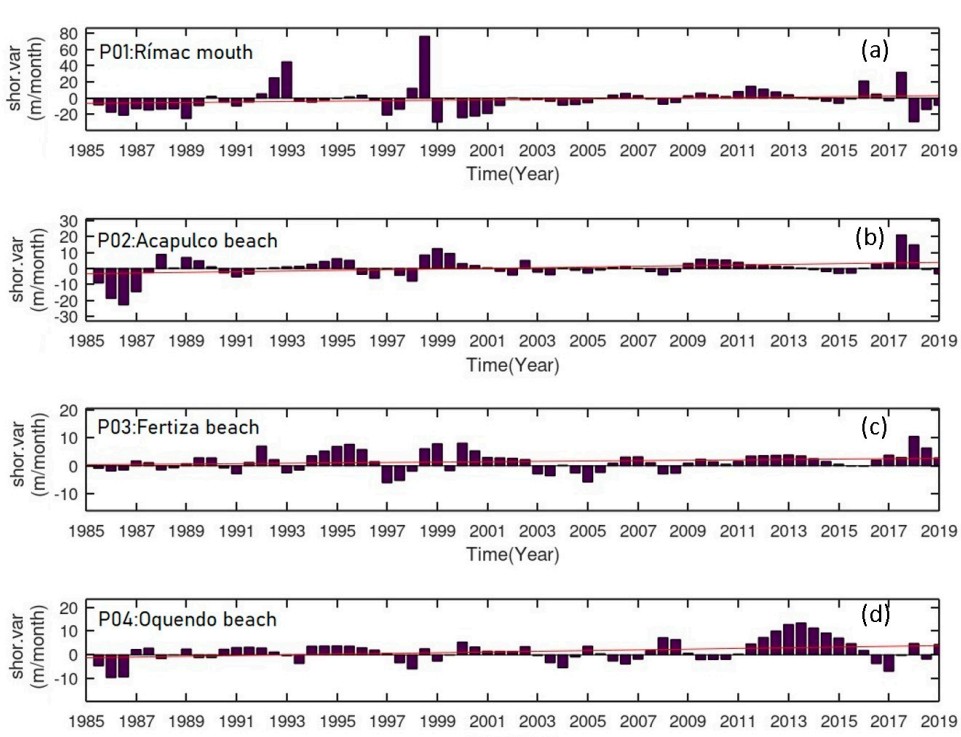

**Figure 14.** Average erosion and accretion rate in control points in m/month, (**a**) P01: Rimac mouth, (**b**) P02: Acapulco beach, (**c**) P03: Fertiza beach, (**d**) P04: Oquendo beach.

Shorelines variation rates were also analyzed every five years to calculate the average rate of shoreline variations for both EN events and no EN events (Table 5). The results clearly show that P01 had the highest rates of shoreline variation across all years studied, particularly in the period between 1995 and 2000, when one of the strongest EN events (1997–1998) produced an average erosion rate of 29.44 m/month. This rate approached the highest reported erosion rate caused by the strong EN of 1982–1983 in the years between 1985 and 1990. However, this EN of 1982–1983 was not included in this analysis due to the lack of satellite imagery for that period. Overall, EN years had much higher erosion and accretion rates when compared to normal years (Figure 15).

**Table 5.** Average coastline accretion/erosion in meters/month between shorelines control points in normal periods and EN events. The highest values for each period are highlighted in bold.

| Loc. | Period 1985–1990 | | 1990–1995 | | 1995–2000 | | 2000–2005 | | 2005–2010 | | 2010–2015 | | 2015–2019 | |
|------|--------|--------|--------|--------|--------|--------|--------|--------|--------|--------|--------|--------|--------|--------|
|      | Normal | Niño | Normal | Niño | Normal | Niño | Normal | Niño | Normal | Niño | Normal | Niño | Normal | Niño |
| P01 | −13.69 | −14.01 | 4.63 | 17.89 | 2.94 | 29.44 | −9.19 | −5.33 | −1.29 | 0.43 | 5.91 | 7.06 | −0.91 | 2.61 |
| P02 | −4.66 | −8.6 | 0.01 | 0.8 | 1.12 | 2.226 | 0.8 | −1.12 | −0.77 | 0.07 | 2.76 | 3.73 | 3.06 | 5.50 |
| P03 | 0.02 | 1.2 | 1.47 | 1.31 | 2.55 | 3.75 | 1.65 | −1.34 | −0.69 | 1.28 | 2.76 | 3.12 | 3.10 | 3.75 |
| P04 | −1.91 | 2.84 | 1.49 | −0.01 | 0.41 | 0.14 | 0.73 | −2.94 | 1.10 | −2.96 | 5.39 | 3.98 | 1.93 | 0.01 |

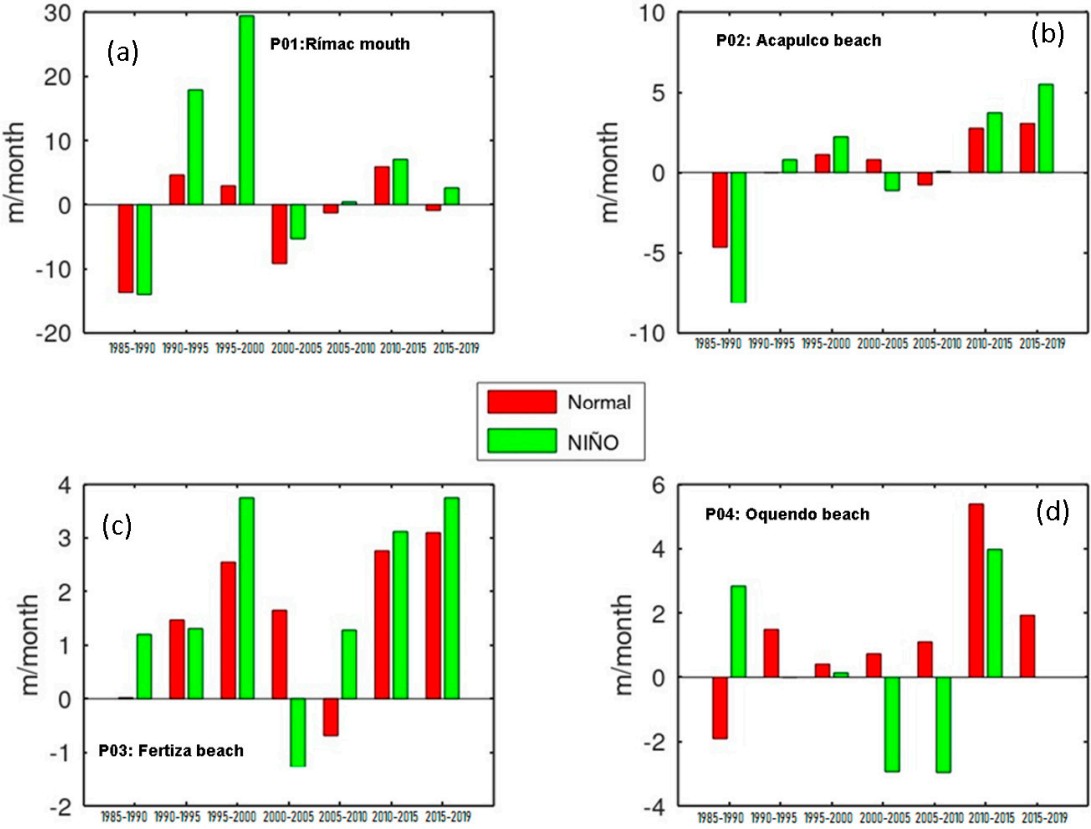

**Figure 15.** Average erosion and accretion rate in control points. Red bar: normal year and green bar: El Niño, (**a**) P01: Rimac mouth, (**b**) P02: Acapulco beach, (**c**) P03: Fertiza beach, (**d**) P04: Oquendo beach

The results show that during the EN events, the extreme changes in sediment discharge from the Rimac River to the Callao Bay dominated the coastline evolution of the Bay for a few years after the event. This discharge effect of EN seemed to be much more important than the slight changes of the wave climate that also occurred during an EN. Clearly, the general long-term accretion trends along the coastline of the Callao Bay were mainly due to the sediment discharge during an EN event, and thus Callao Bay was benefiting from the sediment input of big EN events to sustain its coastline. These results are in accordance with other studies that found that SSY in the Peruvian Andes increases significantly during extreme EN events [24], and in the case of the Rimac River, satellite image analysis of the shoreline suggest that this increased SSY is eventually deposited along the Callao Bay.

These trends of accretion in the Callao Bay due to EN events seem to be opposite of the severe erosion that occurs in U.S beaches [4–8]. In the documented cases in the U.S., the main effect of an extreme EN is changes in storm intensity and water level at the shoreline, while in the case of Callao Bay, the most important effect is the increased sediment input from the Rimac River.

When compared to the Rimac River, the rivers in the Asia–Pacific Region have an opposite behavior, discharging less sediment during EN years than in normal years [29]. Rivers located on the eastern side of the Americas or southwest Africa, where EN events cause less precipitation than in normal years [30], also have lower sediment flux when compared to the Rimac River.

## 5. Conclusions

Shoreline variation of the Callao Bay was analyzed from 1985 to 2019, to describe the main influence of EN on coastline evolution. The major shoreline variations occurred during the extraordinary EN events of 1982–1983, 1997–1998, and 2017. In all three cases a significant delta, 1700–1800 m long and 750–950 m wide, formed at the mouth of the Rimac River. After EN events, the newly developed delta started to erode, and wave-driven currents transported the sediment northward, increasing the

width of beaches located as far as 7 km to the north of the river mouth for 2–3 years after the EN event. During an EN, rates of erosion and sedimentation increased over the normal rates of no EN years.

The results show that during EN events, the extreme growth of the Rimac River led to a bigger sediment supply delivered to the Callao Bay, and it dominated the coastline evolution of the bay for a few years after the event. This EN discharge effect seemed to be much more significant for the evolution of the coastline of the Callao Bay than the slight changes of the wave climate that also occur during an EN. Clearly, the long-term accretion trends along the coastline of the Callao Bay are mainly due to the sediment discharge during the EN event, thus Callao Bay benefits from the sediment input of big EN events to sustain its coastline.

**Author Contributions:** E.G. and C.R. conceptualized the idea EG managed the project, E.G., C.R., and A.D. contributed to the analyses and writing the manuscript; A.D edited the final draft. All authors have read and agreed to the published version of the manuscript.

**Funding:** This research was funded by Universidad Peruana de Ciencias Aplicadas under Project 128: A-099-2019.

**Acknowledgments:** The authors would like to thank to Universidad Peruana de Ciencias Aplicadas.

**Conflicts of Interest:** The authors declare no conflict of interest. The funders had no role in the design of the study; in the collection, analyses, or interpretation of data; in the writing of the manuscript, or in the decision to publish the results.

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
