# Peer review of "Influence of the El Niño Phenomenon on Shoreline Evolution. Case Study: Callao Bay, Perú"

_jmse, doi:10.3390/jmse8020090_

Round 1

Reviewer 1 Report

The manuscript “Influence of El Niño Phenomenon on Shoreline Evolution. Case study: Callao Bay - Perú” by E. Guzman, C. Ramos and A. Dastgheib attempts to define the recent evolution of the Rimac River mouth based on the recurrence of El Nino events in the last 35 years. As sediment redistribution is paramount when it comes to shoreline accretion and erosion, I see merit in the paper, although I’m not sure the present form is that accurate to deserve immediate publication. In the following comments I raise some observations the Authors must address to improve the manuscript. Aerial imagery is a marvelous mean to approach coastline evolution, but it should always be associated to fieldwork surveys to take the most out of it. In my opinion, major revisions are needed before acceptance.

Main Comments

Case Study: I suggest providing more details about the average wave state and the tidal regime in the area as well as sediment grain-size. They are essential in discussing the results.

Case Study: a description of the effect of port structures (if any) on coastal morphodynamics would be useful to better understand sediment distribution.

Materials and Methods: I’m not sure the resolution of 15 and 30 m is sufficient to support the interpretations of the results. For instance, Fig. 13 shows yearly average rates that sometimes are well within that resolution. In this sense in-situ topographic surveys would have been a nice back-up to the trends the Authors pointed out.

Results: no sediment grain-size assessment was made. Details about grain-size distribution along the shoreline would have been a great addition to strengthen the Conclusions.

Discussion: I would expect a comparison between what the Authors described at the Rimac River mouth to some other study cases in literature. Is this a trend pointed out only at Callao Bay? Was it ever described elsewhere?

Conclusions: here the Authors didn’t really explain the results of the research, summarizing the most important findings and highlighting potential future perspectives, which is something I would expect from the concluding section.

Conclusions: text lines 243-249 are identical to text lines 229-234 in the Results and Discussion section. Please rephrase. English: I suggest a thorough English editing as there are several typos.

Additional comments are provided in the attached PDF document.

Reviewer 2 Report

The manuscript submitted by E. Gunzman et al. concerns the possible effects of El Nino (EN) events on a shoreline adjacent to a river mouth in Peru. The topic of the paper is original because most of the studies conducted on the impacts of EN events on coasts were mainly concerned with beach, dunes or cliff shorelines rather than river mouth shorelines or river-influenced shorelines. The work is based on the use of LANDSAT satellite images from 1985 to 2019, which allowed to measure shoreline changes at a relatively high frequency (yearly measurements to several measurements during a single year).

Although some interesting information is provided, the manuscript suffers from several shortcomings (see below). The scope of the paper is also somewhat limited as it concerns relatively localized effects of EN events on the coast and is of local interest. Most importantly, what is strongly lacking in the manuscript is a real Discussion of the results and their significance in which the findings of this study would be compared with other studies that were previously published on the effects of El Nino events on shorelines.

The English language is generally correct, but there are some grammatical errors and typos that need to be corrected (for example: “waves driven currents” instead of “wave driven currents”, “the results shows” instead of “the results show”,…).

SPECIFIC COMMENTS:

- In the Abstract, the authors state that “the effects of EN… on coastal zones and especially in evolution of coastlines is not addressed as much”. This may be true for river-influenced shorelines, but I don’t think this is the case for all shorelines, because there is a significant body of literature concerning the impacts of El Nino events on shoreline changes (see examples below) which are scarcely referred to in the manuscript.

Among others, the list of papers below attest the interest of the scientific community for the impacts of El Nino on the coast:

Allan, J.C., Komar, P.D., 2002. Extreme storms on the Pacific northwest coast during the 1997–8 El Nino and 1998– 9 La Nina. Journal of Coastal Research, 18, 175-193.

Barnard et al., 2015. Coastal vulnerability across the Pacific dominated by El Niño/Southern Oscillation. Nature Geoscience, 8(10), 801-807.

Barnard et al., 2017. Extreme oceanographic forcing and coastal response due to the 2015–2016 El Nino. Nature Communications, 8: 14365.

Dingler, J.R., Reiss, T.E., 2001. Changes to Monterey Bay beaches from the end of the 1982–83 El Nino through the 1997– 98 El Nino. Marine Geology, 18, 249-263.

Kaminsky, G. M., Ruggiero, P., Gelfenbaum, G. R., 1998. Monitoring coastal change in southwest Washington and northwest Oregon during the 1997/98 El Nino. Shore & Beach, 66, 42-51.

Sallenger, A. H. et al., 2002. Sea‐cliff erosion as a function of beach changes and extreme wave runup during the 1997–1998 El Nino. Marine Geology, 187, 279-297.

Storlazzi, C.D., Willis, C.M., Griggs, G.B., 2000. Comparative impacts of the 1982– 3 and 1997–8 El Nino winters on the central California coast. Journal of Coastal Research, 16, 1022-1036.

Young et al., 2018. Southern California Coastal Response to the 2015–2016 El Niño. Journal of Geophysical Research: Earth Surface, 123. https://doi.org/10.1029/2018JF00477

- lines 67-68: the authors say that “two extraordinary EN… occurred in 1982-83 and 1997-98 and had high repercussion on social and economic activities”. But this was said already stated before (line 54) when the authors write that “highest NINO events occurred during 1982/83 and 1997/98”. Unnecessary repetition.

Section 2. Case Study:

- Some information is given in this section concerning the discharge of Rimac river, but more information should be given on the study area in this section, notably the hydro-meteorological forcings. Nothing is said about the tidal range, wind and wave regimes (wave heights? Wave directions? etc…).  

- line 76: The authors refer to” Callao Bay located in central coast of Peru (Figure 2). But there is no indication of Callao Bay in Figure 2.

- An arrow points to “River discharge” in Figure 2. I think the term “River outlet” would be more appropriate.

- lines 81-82: The authors mention that the Rimac River is the main source of sediment to the bay, but no information is given about the sediment load. Are there any measurements of sediment load on this river? What types of sediment (proportion of sand, silt,…)?

- lines 97-99: It is stated that the “Rímac River provides higher sediment supply during summer seasons between January and May and especially during some EN events when discharge increase significantly respect to normal conditions”. How much higher is the sediment supply during these conditions? On what is based this statement? Any measurements of sediment load?

- lines 100-102: The authors give an estimation of 3.4 million cubic meters of sand that was presumably supplied by the Rimac River during 2017 El Nino Costero, this sediment volume having been estimated from bathymetry changes between 2017 and 2018. However, it is not known what area was covered by the bathymetry surveys. Was it restricted to the river mouth? If the survey covered a larger area offshore, it cannot be totally ruled out that some of the sediment that accumulated between these bathymetry surveys may come from another source (i.e., not from the Rimac river).

Section 3. Materials and Methods: Some useful information is given in this section concerning the data used for conducting this study and data processing, but no error margin is given (which is different from image resolution) and it is therefore not possible to assess the accuracy of shoreline change.

- line 144: It is written that “Satellite image shows total variation in shoreline which include effect of climate changes…”. I don’t understand what the authors mean by that.

Section 4. Results and Discussions (as mentioned above there is no real Discussion of the results in this section, or in any other section of the paper). Some information about the forcing conditions before and during satellite acquisition is missing and should be provided for a better understanding of the mechanisms that may have been responsible for the observed shoreline changes (see below):

- lines 155-157: It is stated that a delta that was visible in May 1998 (after EN 1997-98) has been partly eroded by December 1998. Some erosion of the delta apparently occurred, but how can that be explained? In response to what forcings/processes? Just saying that it was eroded by waves and currents is too imprecise. Any storm between May and December 1998, high water levels, high energy waves?... Some information about what could explain this erosion should be given.

- Figure 6: The December 1998 image is of poor quality that makes difficult to assess the actual morphology of the river-mouth delta at that date. The author claim that erosion occurred between May and December 1998 based on these two images, but can the apparent changes in shoreline position could be also due to differences in water level between these two dates (due to tide or to a positive or negative surge)? Information should be given about the water level at the time of image acquisition which should enable to estimate how much of the observed change in shoreline position could be due to differences in water level.

- lines 157-158: it is written that in December 1998 the “Rímac river delta still exists at the river mouth but is eroded by littoral transport dynamics…”. A delta cannot be eroded by “littoral transport dynamics”; it can be eroded by waves or may be eroding due to a loss of sediment. This should be rephrased.

- lines 157-158: the authors talk about “littoral transport dynamics driven by waves and currents”. This is very vague. What currents? Longshore wave-driven currents? Tidal currents? Other currents?...

- lines 158-159: it is written that “Similar conditions occurred in 2017 during “El Nino Costero”. What conditions? Similar to what? To the conditions that favored the formation of the delta that was visible in May 1998?

- line 161: the authors write that the “delta has a northward movement mainly due to wave driven sediment transport in Callao bay”. The main littoral drift is therefore presumably directed northward, but this should have explained before (in section “2. Case Study”).

- Figures 8 and 9: the colors used in the figures for showing the shoreline positions at different dates are confusing. In Figure 8, the yellow line corresponds to the first date and the red line to the second date. But in Figure 9, it is the opposite: the yellow line corresponds to the last date and the red line corresponds to the first date. The same color should be used for the first (or the last) date.  

- lines 194-195: based on the results shown in Figures 11 and 12, the authors assume that to the north of the river mouth, the “coastline has accreted few months after EN events”. According to these figures, it looks like accretion on these beaches (especially Oquendo beach) occurred several years after EN events (not after a few months).

- Figure 13: 4 graphs are shown in Figure 13. I assume that each graph corresponds to a control point. But this is not indicated in the figure.

- Table 4: Why are some numbers highlighted in red in the table?

- lines 232-233: The authors claim that “the general long-term accretion trends along the coastline of the Callao Bay are mainly due to the sediment discharge (from the Rimac River?) during the EN event”. However, although there results suggest that this is probably the case, there are no data concerning sediment discharge from the river for supporting this hypothesis.

Round 2

Reviewer 1 Report

No further comments.

Author Response

Thanks very much for the suggestion. The manuscript was revised for a professional English editor

Reviewer 2 Report

The authors made appropriate changes to the originally submitted manuscript and give satisfactory explanations in their “Reply to reviewer”. However, the text of the revised manuscript still contains grammatical errors that need to be corrected. Below are some examples:

- Line 80: “TP-Callao has two breakwaters with approximately 1.3 km of length”. Rewrite: “TP-Callao has two breakwaters approximately 1.3 km long”.

 Line 100: “There’s very few information…”. Should be written “There is very few…”.

- Lines 124-125: “During this events, 82% to 97% of SSY occurs between January and April, which coincide with summer season in south hemisphere”. Should be written “During these events…”, “… which coincides…”, “… with the summer season in the southern hemisphere”.

- Line 130: “…between January to march”. “march” should be written “March”.

-Line 194: “Similar conditions of delta formation were occurred in 2017…”. This should be written “…delta formation occurred in 2017”.

- Line 227: “The Deep-water wave climate…”. Should be written “The deep-water…”.

- Lines 233-234: “Wave direction (Figure 12c) doesn’t have significant variation in deep-waters…”. “Doesn’t” should be written “does not”. I do not understand what “variations in deep-waters” mean. Variations in deep-water directions?

- Lines 235-237: “These values are much less comparing extreme wave conditions under EN in some other locations around the world, for instance in California beaches major storms due EN generated wave height greater than 10m [7]”. “comparing” should be changed to “compared to”. The rest of the sentence is awkward and should be rephrased. “in California beaches” should be changed to “along California beaches”. Moreover, it is not possible to observe wave heights higher than 10 m (even on California beaches); this is probably offshore.

- Lines 286-287, where it is written that “… extreme changes in the sediment discharge from 286 the Rímac River to the Callao Bay dominates (sic) the coastline evolution…”. “Dominates” should be written “dominate”.

- Line 292-293: “… increases significantly due extreme EN events…”. Due to?

- Line 298: “…the most important effect is the amplified sediment input…”. Wrong phrasing; “amplified sediment input” is incorrect. “Increased sediment input” would be more appropriate.

- Line 305: “Shoreline variations analyses in Callao bay is performed…”. This phrase is incorrect: “Shoreline variations analyses “are” (or “were”) performed” (not “is”).

- Lines 306-307: “Main significant variation in shoreline were occurred in extraordinary EN events…”. “were occurred in” is incorrect. Should be changed for something like “…variation in shoreline resulted in…”

There are several other errors in the text which still needs to be reviewed and edited by a native English translator/editor.
